# EVGAP: Egocentric-Exocentric Video Groups Alignment Pre-training

## Abstract

Aligning egocentric and exocentric videos facilitates the learning of view-invariant features, which significantly contributes to video understanding. While previous approaches have primarily focused on aligning individual ego-exo video pairs, our method extends this concept by aligning groups of synchronized egocentric and exocentric videos. This strategy enables the model to capture more comprehensive cross-view relationships across densely captured viewpoints, enhancing its capacity for robust multi-view understanding. Therefore, we develop a pipeline based on contrastive learning for **E**gocentric-exocentric **V**ideo **G**roups **A**lignment **P**re-training (EVGAP). Our method introduces several key innovations: 1) a novel video pre-training paradigm that extends alignment from ego-exo video pairs to ego-exo video group alignments; 2) an innovative two-step training process that leverages the abundant ego-exo video pair data to support the learning of ego-exo video group alignments, transitioning from sparse to dense viewpoints; and 3) the application of auxiliary losses to progressively align videos from different perspectives. Extensive ablations illustrate the effectiveness of our approach in single-view and multi-view downstream tasks. We also find that our approach facilitates the tasks inluding novel views. The codes will be available upon acceptance.

## 1 Introduction

Since the majority of available video data is exocentric, models designed for exocentric video understanding benefit from extensive large-scale datasets. Moreover, research has shown that exocentric video learning can also facilitate egocentric video understanding (Li et al., 2021b). Consequently, a significant research focus on learning view-invariant video features to align egocentric and exocentric videos, thereby enhancing the understanding of both perspectives. Among them, they aligning ego-exo[1] video pairs that share the same or similar semantics, depending on paired (Sigurdsson et al., 2018a; Ardeshir & Borji, 2018; Sermanet et al., 2018; Yu et al., 2019; 2020) or unpaired views (Xue & Grauman, 2023; Wang et al., 2023). Based on egocentric and exocentric video data (Sigurdsson et al., 2018b; Sener et al., 2022; Grauman et al., 2024), the resulting joint feature space facilitates a range of tasks, such action recognition (Kazakos et al., 2019; Yonetani et al., 2016), action anticipation (Furnari & Farinella, 2020; Abu Farha et al., 2018), video summarization (Lee & Grauman, 2015; Del Molino et al., 2016), robot learning (Bharadhwaj et al., 2023) and so on.

The existing methods predominantly focus on one-to-one pairing to learn view-invariant features, where each egocentric video is aligned with a single exocentric video, as illustrated in Figure 1 (a). On the other hand, videos can also be grouped based on unpaired video grouping via language (Wang et al., 2023) or paired video grouping via dense synchronized views (Sener et al., 2022). Thus inspired, we apply such grouping for both ego and exo to form ego video groups and exo video groups, as illustrated in the top and bottom of Figure 1 (b), respectively. Under this setting, we empirically find that the model potentially has suboptimal performance when we simplify the video group alignment by using the one-to-one pairwise alignment.

**Specifically, we propose a novel alignment between ego video groups and exo video groups.** To our best knowledge, this is the first attempt to conduct grouped video pre-training for ego-exo

---

[1]For simplicity, the terms 'ego' and 'exo' are used for 'egocentric' and 'exocentric', respectively.

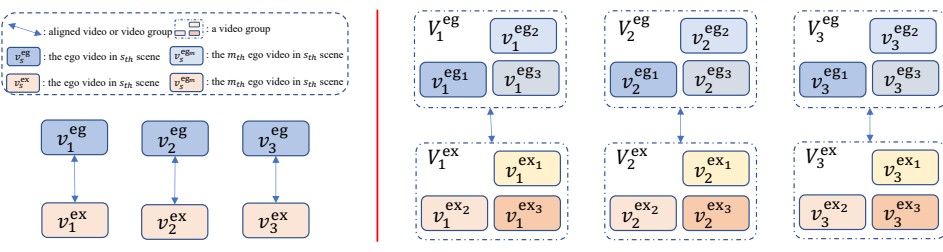

(a) Align ego and exo videos          (b) Align ego and exo video groups

Figure 1: Alignment between (a) ego-exo video pairs and (b) ego-exo video groups. The subscript represents the scene index, and all videos from either the egocentric or exocentric perspectives within the same scene are synchronized. In ego-exo video group alignment, each ego or exo video group within a scene includes multiple viewpoints. Unlike aligning individual ego-exo pairs in method (a), method (b) aligns ego-exo video groups.

alignment. In particular, we work on paired video groups where we have dense[2] synchronized views. This new problem raises the following questions: **(1)** How can we optimally leverage ego-exo video groups to improve multi-view video representation learning? As outlined above, the ego-exo groups can also be considered as a collection of individual ego-exo pairs. **(2)** Compared with aligning several ego-exo video pairs, is aligning ego-exo video groups more effective? Furthermore, ego-exo video group data with dense viewpoints is relatively scarce (Sener et al., 2022), while a significant portion of existing ego-exo datasets comprises one-to-one video pairs. **(3)** How can we leverage these ego-exo pairs data in our model to facilitate the alignment of ego-exo video groups?

To answer question (1), all exo and ego video groups are processed through a shared video encoder to establish a similarity metric between the ego and exo perspectives based on the feature outputs, with a contrastive loss (Radford et al., 2021) applied to this metric. The answer to question (3) arises from our proposed two-step pretraining strategy, aimed at learning a view-invariant representation of multi-view ego and exo videos. In the first training step, video data is assigned to ego-exo pairs. This facilitates training on large-scale data with sparse viewpoints and limited-scale data with dense viewpoints. In the second step, the model weights learned from the first step are used for initialization, and only ego-exo video groups are adopted. In this step, we apply dense contrastive learning to handle alignment of video groups efficiently. Furthermore, unlike multi-modal contrastive learning (Radford et al., 2021; Caron et al., 2021; Jia et al., 2021), where the contrastive loss is typically applied at the final output of the encoders, in the visual domain we introduce an auxiliary loss. Specifically, we obtain contrastive losses after each layer of our model in both steps. This approach facilitates more efficient convergence.

To answer the question (2), we finetune our pre-training model on Assembly101 for fine-grained video understanding tasks, *e.g.* action segmentation and action anticipation. We conducted extensive ablations which have verified the effectiveness of the proposed two-step pretraining strategy. Our approach improves both action segmentation and anticipation by 0.9%, and 1.9% on average. Interestingly, our method outperforms the baseline by almost 2.3% and 2.5% in the novel view setting of TAS and action anticipation, which shows the potential for learning comprehensive view-invariant representation.

## 2 RELATED WORK

### 2.1 EGO-EXOCENTRIC VIDEO ALIGNMENT

Aligning egocentric and exocentric views for feature learning is a challenging task that has been approached from various perspectives. One line of work focuses on joint attention mechanisms to co-analyze spatial-temporal relationships across both views, improving feature alignment for tasks like action recognition and object interaction (Yu et al., 2019; Sigurdsson et al., 2018a; Yu et al., 2020). By minimizing the distance between corresponding frames and maximizing the distance between

---

[2]Dense view means there are multiple cameras for both egocentric and exocentric views.

non-corresponding ones, recent works have shown significant improvements in view-invariant representation learning via contrastive learning (Xue & Grauman, 2023; Qian et al., 2021). The studies in (Sudhakaran et al., 2019; Yu et al., 2019; Ji et al., 2021) adapt attention mechanisms, to selectively focus on important features in both views, further refine the alignment of spatial-temporal information. Recent works have explored cross-view image synthesis and bridging the domain gap between different perspectives using generative adversarial networks (GANs) (Elfeki et al., 2018; Regmi & Borji, 2018; Regmi & Shah, 2019; Liu et al., 2020; 2021) or diffusion model (Luo et al., 2024). Lastly, multimodal fusion techniques and multi-view learning methods have also been applied to this problem. For example, cross-view fusion models improve recognition tasks by integrating pose and action data from both perspectives (Iskakov et al., 2019).

## 2.2 CONTRASTIVE LEARNING

Contrastive learning (Radford et al., 2021; Chen et al., 2020; He et al., 2020; Caron et al., 2020) has emerged as a powerful paradigm in self-supervised representation learning, particularly in tasks involving cross-modal alignment and multi-view learning. A prominent example in contrastive learning is CLIP (Radford et al., 2021) which aligns images with their corresponding text descriptions using contrastive loss, leveraging large-scale natural language supervision to enable zero-shot learning across diverse visual tasks. BiLIP (Li et al., 2022) further refines contrastive learning by integrating bidirectional modeling. Moreover, ALIGN (Jia et al., 2021) scales multimodal representation learning using noisy text descriptions in a contrastive framework. FLAVA (Singh et al., 2022) and UniCL (Yang et al., 2022) further unify vision and language modalities, enhancing performance in tasks like image retrieval and captioning. ALBEF (Li et al., 2021a) extends this further by proposing a method that aligns image and text representations before fusing them, achieving strong results across various vision-language benchmarks. PixPro (Xie et al., 2021) focuses on fine-grained, pixel-level representations for dense prediction tasks using a local contrastive learning approach. Additionally, ReLICv2 (Tomasev et al., 2022) extends contrastive learning by incorporating relational inductive biases, improving contextual understanding in relational reasoning and object detection tasks. Other notable methods include SimCLR (Chen et al., 2020) and MoCo (He et al., 2020), which focus on learning visual representations from images by contrasting positive and negative pairs.

## 3 METHOD

In this section, we present the Ego-Exo Video Group Alignment Pretraining (EVGAP) and start with the encoder structure in Sec.3.1. Then, for detailed training procedure, the first step for exo-ego perspective pairs and the second step for ego-exo video group pairs are introduced respectively in Sec.3.2 and Sec.3.3, followed by the auxiliary loss in Sec.3.4.

### 3.1 EGO-EXO VIDEO GROUP ALIGNMENT PRETRAINING

Building on the success of contrastive learning on multi-task generalization and zero-shot capabilities via aligning data from multi-modality, we adopt a contrastive pre-training framework within the visual domain using multi-view video data. We propose Ego-Exo Video Group Alignment Pretraining (EVGAP), designed to learn a shared feature space for video frames from multiple perspectives, to enhance performance of downstream video analysis tasks and explore the zero-shot capacity of novel views.

**Ego-Exo Video Groups Alignment.** EVGAP is designed to learn joint representations from video group data captured across multiple synchronized viewpoints. This process involves aligning video clips between egocentric (ego) and exocentric (exo) perspectives, enabling multi-view videos with dense perspectives to be projected into a shared feature space. The input for alignment consists of sets of video sequences, combining both egocentric and exocentric video groups. Specifically, given $S$ scenes, for the $s^{th}$ scene, we capture $M$ egocentric (first-person) perspectives to form the ego video group $V_s^{\text{eg}} = \{v_s^{\text{eg}_m}\}_{m=1}^M$, and $N$ exocentric (third-person) perspectives to form the exo video group $V_s^{\text{ex}} = \{v_s^{\text{ex}_n}\}_{n=1}^N$, all synchronized. Here, $s$ denotes the scene index, while $m$ and $n$ represent the indices of the ego and exo perspectives, respectively. The ultimate goal of EVGAP is to align and pair the ego-exo video groups $(V_s^{\text{eg}}, V_s^{\text{ex}})$ across all scenes. To account for the dense prediction

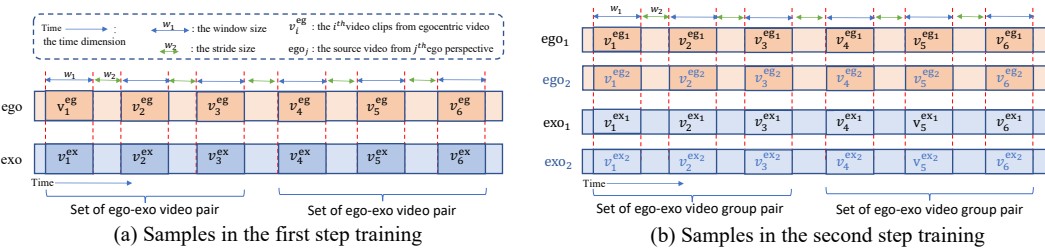

(a) Samples in the first step training       (b) Samples in the second step training

Figure 2: Strategies to build the batch data in training for two steps. In (a), the example given a source ego and a source exo video, video clips are sample by given window size and stride. The video clips in video clip set will appear in the same batch. In (b), the example involves two egocentric and two exocentric source videos, from which video clips are sampled similarly to (a), resulting in a video group composed of synchronized video clips.

of downstream tasks , we sample multiple consecutive frames from videos, to construct clip-based $v_s^{\text{eg}_m}$ and $v_s^{\text{ex}_n}$.

**Data Batch.** As outlined in Sections 3.2 and 3.3, the pretraining process consists of two distinct steps, each utilizing different input data pairs. Furthermore, the model relies on batch-based contrastive learning, making the selection of negative samples and the balance between positive and negative sample ratios crucial for effective training in both steps.

In the first step, the model is fed with ego-exo video pairs for alignment. In the second step, however, ego-exo video group pairs are used for alignment. In both steps, video clips are sampled from the source videos to construct the pairs. The key difference is that in the second step, clips are sampled from ego-exo video groups, such as $(\{\text{eg}_1, \text{eg}_2\}, \{\text{ex}_1, \text{ex}_2\})$, representing two synchronized viewpoints for both egocentric and exocentric perspectives, as illustrated in Fig 2(b). We sample video clips from each source video using a window size of $w_1$ and a stride of $w_2$. Due to synchronization, video clips sampled from the same timestamps represent the same scene, thereby sharing the same semantic content, denoted by the subscript. For the $i^{th}$ clip, we obtain ego-exo video pair $(v_i^{\text{eg}}, v_i^{\text{ex}})$ in the first step, and ego-exo video group pair $(\{v_i^{\text{eg}_1}, v_i^{\text{eg}_2}\}, \{v_i^{\text{ex}_1}, v_i^{\text{ex}_2}\})$ in the second step. Since we sample multiple clips for a video, here we replace $s^{th}$ by $i^{th}$.

Since contrastive learning relies on the distance between paired samples within a batch, selecting appropriate batch samples is crucial. When the samples in a batch are distinct, it becomes easier for the model to pair the synchronized pairs, which we refer to as the *easy case*. In this case, the model may exhibit a form of *lazy learning*, failing to capture the alignment we want between the views. Conversely, when the samples in a batch are highly similar, the task of pairing becomes significantly more challenging, referred to as the *difficult case*. An example of this occurs when more temporally adjacent clips from the same video are selected, resulting in highly similar features. In such cases, the model struggles to differentiate between the pairs, leading to poor convergence and limiting its ability to learn meaningful alignments. In the first step, we include "sets of ego-exo video pairs" in each batch, whereas in the second step, "sets of ego-exo video group pairs" are used in each batch.

**Visual Encoder.** For the visual encoder $H(\cdot)$, we construct the architecture by stacking $L$ Transformer encoder layers, applying layer normalization after each layer. All videos share the same encoder to extract output features, with the class token serving as the feature representation for each video clip. Unlike the VLP paradigm, where distinct encoders are used for images and text, we employ a unified encoder in EVGAP for all videos from both egocentric and exocentric views, as they represent similar visual signals. The class tokens are then fed into the loss functions to facilitate different stages of pretraining.

### 3.2 FIRST STEP FOR EGO-EXO VIDEO PAIR ALIGNMENT

Comparing the relation matrices of ego-exo video alignment and ego-exo video group alignment, the latter aims to capture additional group-level information beyond what is represented in the former. Moreover, the number of ego-exo video pairs is significantly bigger than that of ego-exo video group

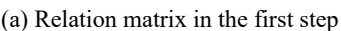

| | (a) Relation matrix in the first step | (b) Relation matrix in the second step |

Figure 3: (a) and (b) represent the relation matrices for the first and second steps of training, respectively, for a given batch of data. The first row of (a) and the second row of (b) represent the ego videos, whereas the first column of (a) and the second column of (b) represent the exo videos. The first row and column of (b) indicate the groups including ego or exo videos with the same semantic content. A value of 1 in the matrix indicates that the ego and exo videos share the same scene. In (a), the alignment represents ego-exo video pairs, resulting in a diagonal entirely composed of ones. In (b), the alignment involves ego-exo video group pairs, where multiple ego and exo videos within a group share the same scene. Consequently, the diagonal contains several blocks with all ones, reflecting these group-wise associations.

pairs. Therefore, we leverage the ego-exo video pair alignment to pre-train model weights, which are then used as the initialization for group alignment learning in the second step.

Specifically, in the first step, we aggregate a large number of ego-exo video pairs to train a general model for ego-exo video alignment. Given a batch of paired video features comprising $B$ scenes: $\{(v_i^{\text{eg}}, v_i^{\text{ex}})\}_{i=1}^{B}$, the relation matrix for the batch, as illustrated in Fig. 3(a), assigns a value of '1' to indicate that the corresponding ego and exo videos originate from the same scene and should be aligned. This implies that $(v_i^{\text{eg}}, v_i^{\text{ex}})$ are positive pairs, whereas $(v_i^{\text{eg}}, v_j^{\text{ex}})$, where $i \neq j$, are not. Based on the ego and exo perspectives, the batch can also be divided into an ego video set $V^{\text{eg}} = \{v_i^{\text{eg}}\}_{i=1}^{B}$ and an exo video set $V^{\text{ex}} = \{v_i^{ex}\}_{i=1}^{B}$. All videos, including both ego and exo perspectives, are fed through the visual encoder $H(*)$ to produce the output feature embeddings $H(V^{\text{eg}}) = \{H(v_i^{\text{eg}})\}_{i=1}^{B}$ and $H(V^{\text{ex}}) = \{H(v_i^{\text{ex}})\}_{i=1}^{B}$. To extend the contrastive loss from CLIP to ego-exo alignment, the loss function is formulated as follows:

$$\mathcal{L}_{\text{pair}}(V^{\text{eg}}, V^{\text{ex}}) = -\frac{1}{2B} \sum_{i=1}^{B} \left( \log \frac{e^{t \cdot H(v_i^{\text{eg}})^{\top} H(v_i^{\text{ex}})}}{\sum_{j=1}^{B} e^{t \cdot H(v_i^{\text{eg}})^{\top} H(v_j^{\text{ex}})}} + \log \frac{e^{t \cdot H(v_i^{\text{eg}})^{\top} H(v_i^{\text{ex}})}}{\sum_{j=1}^{B} e^{t \cdot H(v_j^{\text{eg}})^{\top} H(v_i^{\text{ex}})}} \right) \quad (1)$$

where $t$ is the temperature parameter.

This formulation encourages the model to maximize the similarity between matched video features $(H(v_i^{\text{eg}}), H(v_i^{\text{ex}}))$ across views while minimizing the similarity between mismatched ones.

### 3.3 SECOND STEP FOR EGO-EXO VIDEO GROUP ALIGNMENT

This section focuses on aligning ego-exo video groups to capture additional relationships within the egocentric perspectives and within the exocentric perspectives. To achieve robust performance given the limited training data, we utilize the learned weights from the first step to initialize the model for this stage. It is assumed that the first step has effectively captured pairwise ego-exo video alignment, and these pre-trained weights will enable the model to refine its understanding further by concentrating on group-level alignment.

In the second step, for batch data, the inputs are $B$ ego-exo video group pairs, denoted as $\{(V_i^{\text{eg}}, V_i^{\text{ex}})\}_{i=1}^{B}$. As illustrated in Fig. 3(b), we have an egocentric video from the $i^{th}$ scene, $v_i^{\text{eg}_m}$, its positive samples include all exocentric videos from the same scene, denoted as $V_i^{\text{ex}} = \{v_i^{\text{ex}_n}\}_{n=1}^{N}$.

Negative samples are exocentric videos from different scenes, $V_j^{\text{ex}}$, where $i \neq j$. Given that there are $M$ viewpoints for egocentric videos, all videos within the egocentric video group $V_i^{\text{eg}} = \{v_i^{\text{eg}_m}\}_{m=1}^M$ must align with each exocentric video from the same scene.

As we need to supervise $M \times N$ positive samples in a video, we adopt the sigmoid loss used in SigLIP (Zhai et al., 2023). And inorder to keep balance rate between positive and negative samples, we ignore the device communication to not to gather features from cross-devices.

In this context, the modified SigLIP loss (Zhai et al., 2023) for ego-exo group alignment is formulated as follows:

$$\mathcal{L}_{\text{EVGAP}}(V^{\text{eg}}, V^{\text{ex}}) = -\frac{1}{B^2 \times M \times N} \sum_{i=1}^{B} \sum_{j=1}^{B} \sum_{m=1}^{M} \sum_{n=1}^{N} \log \left( \frac{1}{1 + e^{z_{ij}(-t \cdot H(v_i^{\text{eg}_m}) \cdot H(v_j^{\text{ex}_n}))}} \right), \quad (2)$$

where $z_{ij}$ represents the label for a given pair consisting of an egocentric video and an exocentric video. Specifically, $z_{ij} = 1$ if $i = j$, indicating a positive match; otherwise $z_{ij} = -1$ if $i \neq j$, indicating a negative match.

### 3.4 Auxiliary Loss

The image-text pre-training aligns features in the last decoder layer, we assume that different modalities need deep neural networks for representation before alignment. Unlike image-text pairs, the dual views expected to be aligned are both visual signals. Thus, we explore the potential of middle supervision, *i.e.* appending auxiliary losses on the output of each visual encoder layer. In order to represent outputs from different layers, we represent the sub-visual encoder $h_l(\cdot)$ as the model with the previous $l$ layers in the visual encoder $H(\cdot)$, where $l = 1, 2, .., L - 1$.

The auxiliary losses is the sum of the loss of output from $L - 1$ layers. For the $l^{th}$ layer, the loss $\mathcal{L}_{\text{step1}_l}$ and $\mathcal{L}_{\text{step2}_l}$ for the two steps can be derived by replacing $H(*)$ with $h_l(*)$ in $\mathcal{L}_{\text{pair}}$ and $\mathcal{L}_{\text{EVGAP}}$. resulting in $\mathcal{L}_{\text{Aux\_pair}}$ and $\mathcal{L}_{\text{Aux\_EVGAP}}$, respectively. Consequently, the auxiliary loss and total loss for the first and second steps can be formulated as follows:

$$\mathcal{L}_{\text{Aux\_pair}} = \sum_{l=1}^{L-1} (\alpha_l \cdot \mathcal{L}_{\text{step1}_l}) \qquad \mathcal{L}_{\text{Total\_pair}} = \mathcal{L}_{\text{pair}} + \mathcal{L}_{\text{Aux\_pair}} \quad (3)$$

$$\mathcal{L}_{\text{Aux\_group}} = \sum_{l=1}^{L-1} (\beta_l \cdot \mathcal{L}_{\text{step2}_l}) \qquad \mathcal{L}_{\text{Total\_group}} = \mathcal{L}_{\text{EVGAP}} + \mathcal{L}_{\text{Aux\_group}} \quad (4)$$

where $L$ is the number of visual encoder layers, $\alpha_i$ and $\beta_i$ is the loss weight for the $l^{th}$ outputs.

## 4 Experiment

### 4.1 Dataset

We employ two datasets: Assembly101 and Charades-Ego for the first step, while utilizing the multi-view Assembly101 dataset for egocentric and exocentric perspectives in the second step. **Assembly101** (Sener et al., 2022) is a large-scale, multi-view video dataset tailored for action understanding in complex assembly and disassembly tasks. It contains over 1,000 videos captured from a total of 12 different camera angles for each video, including 4 egocentric (first-person) views and 8 exocentric (third-person) views. The dataset features a diverse set of assembly activities performed by different individuals, providing more than 500 hours of footage. We utilize video features extracted from the TSM (Lin et al., 2019) model pretrained on the Assembly101 dataset, similar to the approach in (Sener et al., 2022). **Charades-Ego** (Sigurdsson et al., 2018b) is an extension of the Charades dataset (Sigurdsson et al., 2016), focusing on everyday activities captured simultaneously from paired egocentric and exocentric perspectives. It includes approximately 7,860 videos recorded in natural home environments. Each video features a set of predefined activities, with detailed annotations for actions, temporal boundaries, and object categories. To obtain features similar to those in the Assembly101 dataset, we apply a window size of 8 frames and process them through the TSM model, using weights pretrained on the Assembly101 dataset.

| Input | Sing-view | | | | | | Two-view | | | | | |
|-------|-----------|---|---|---|---|---|----------|---|---|---|---|---|
| **Method** | **F1@{10, 25, 50}** | | | **Edit** | **Acc.** | **Avg.** | **F1@{10, 25, 50}** | | | **Edit** | **Acc.** | **Avg.** |
| Base | 33.0 | 28.7 | 20.6 | 31.5 | 37.7 | 30.3 | 31.5 | 27.4 | 20.0 | 30.7 | 39.0 | 29.7 |
| (a) Base + random | 32.9 | 28.9 | 21.0 | 31.4 | 37.8 | 30.4 | 32.8 | 28.3 | 19.8 | 31.0 | 38.4 | 30.1 |
| $\Delta_{(a)-\text{Base}}$ | -0.1 | +0.2 | +0.4 | -0.1 | +0.1 | +0.1 | +1.2 | +0.9 | -0.2 | +0.3 | +0.6 | +0.4 |
| (b) Base + step1 | 33.0 | 28.9 | 21.7 | 32.1 | 37.8 | 30.6 | 33.1 | 28.6 | 20.0 | 30.9 | 39.3 | 30.6 |
| $\Delta_{(b)-\text{Base}}$ | +0.0 | +0.2 | +1.1 | +0.6 | +0.1 | +0.3 | +1.6 | +1.2 | +0.0 | +0.2 | +0.3 | +0.9 |
| (c) Base + step2 | 33.3 | 29.5 | 21.6 | 32.6 | 38.0 | 31.0 | 33.3 | 29.3 | 21.5 | 31.9 | 40.0 | 31.2 |
| $\Delta_{(c)-\text{Base}}$ | +0.3 | +0.8 | +1.0 | +1.1 | +0.3 | +0.7 | +1.8 | +1.9 | +1.5 | +1.2 | +1.0 | +1.5 |
| (d) Base + step1 + step2 | 33.4 | 29.6 | 22.1 | 32.5 | 38.5 | 31.2 | 33.7 | 29.9 | 21.7 | 32.2 | 40.4 | 31.6 |
| $\Delta_{(d)-\text{Base}}$ | +0.4 | +0.9 | +0.5 | +1.0 | +0.8 | +0.9 | +2.2 | +2.5 | +1.7 | +1.5 | +1.4 | +1.9 |

Table 1: Ablation study on single-view and two-view temporal action segmentation tasks on the Assembly101 dataset. 'Base' represents the C2F-TCN model. In (a), the pretraining model are stacked with C2F-TCN for end-to-end training from scratch. In (b) and (c), the model utilizes weights from pretraining conducted in only the first or second step, respectively. (d) represents the model trained using weights obtained from both pretraining steps.

## 4.2 PRE-TRAINING DETAILS

**Batch data building.** In the first training step, we set the window size to 20 frames and apply augmentation by scaling the window size within a range of $0.5 \sim 2.0$. The stride between windows is set to 100 frames. Each batch contains a set of 10 video clips from a same source video. In the second training step, we maintain the same configuration from the first step to extract clips from the source videos. Additionally, video clips corresponding to the same timestamps are collected into a video group. A video group consists of eight exocentric videos and four egocentric videos.

**Encoder.** The video features, initially of dimension 2048, are extracted from the TSMLin et al. (2019) model, excluding the last linear layer, with fixed weights pretrained on the Assembly101 dataset. These features are then projected onto a feature embedding space of dimension 512. The resulting embeddings are subsequently fed into an encoder comprising six layers of Vision Transformer (ViT (Dosovitskiy et al., 2021)) encoder blocks. Each encoder block utilizes eight attention heads, a feedforward dimension of 2048, and ReLU activation applied following layer normalization.

**Training settings in the first and second step.** In the first training step, the batch size is set to 200, with a learning rate of $5 \times 10^{-4}$, using the Adam (Kingma, 2015) optimizer. In the second step, the batch size is reduced to 64, with a learning rate of $1 \times 10^{-4}$, and the AdamW (Loshchilov, 2019) optimizer is employed. When computing the loss, the logits scale for the first and second steps are initialized to 1 and $\log_{10}(1/0.7)$, respectively. Additionally, the loss weight for the auxiliary loss is set to 0.2 in both steps.

## 4.3 ABLATION STUDY

In this section, we present ablation studies to empirically validate the different components of our pipeline. All evaluation are conducted on validation split of Assembly101 dataset, and the improvement are showed by $\Delta$. Specifically, we employ the downstream task of temporal action segmentation (TAS) to demonstrate the results. We present the F1 scores at overlaps of 10%, 25%, and 50%, along with the edit score, accuracy, and the average of all metric values. For TAS, we select C2F-TCN (Singhania et al., 2021) as the base model. To leverage the alignment model, we integrate the pre-trained encoder with the C2F-TCN model. Additionally, to evaluate the alignment's effectiveness in multi-view input settings, we extend the TAS task from single-view to two-view, providing ablation results for both. In the two-view temporal action segmentation, the input consists of video features from two different viewpoints.

First, we train the end-to-end model from scratch using random weights, as illustrated in Table 1's method (a). The results indicate that, on average, performance improves by 0.1% and 0.4% for

| Method | F1@{10, 25, 50} | | | Edit | Acc. | Avg. |
|---|---|---|---|---|---|---|
| | **Temporal Action Segmentation** | | | | | |
| ASFormer | 33.1 | 28.4 | 20.5 | 31.2 | 37.4 | 30.1 |
| +EVGAP* | 33.4 | 28.9 | 21.0 | 31.7 | 37.8 | 30.5 |
| +EVGAP | 33.6 | 29.2 | 21.6 | 32.0 | 38.4 | 30.9 |
| $\Delta$ | +0.5 | +0.8 | +1.1 | +0.8 | +1.0 | +0.8 |
| C2F-TCN | 33.0 | 28.7 | 20.6 | 31.5 | 37.7 | 30.3 |
| +EVGAP* | 33.2 | 29.2 | 20.9 | 30.9 | 38.0 | 30.44 |
| +EVPG | 33.4 | 29.6 | 22.1 | 32.5 | 38.5 | 31.2 |
| $\Delta$ | +0.4 | +0.9 | +1.5 | +1.0 | +0.8 | +0.9 |
| LTContext | 33.6 | 28.4 | 20.5 | 32.2 | 38.4 | 30.6 |
| +EVGAP* | 33.8 | 29.1 | 21.2 | 32.4 | 39.2 | 31.14 |
| +EVGAP | 34.2 | 29.6 | 22.2 | 32.8 | 39.7 | 31.7 |
| $\Delta$ | +0.6 | +1.2 | +1.7 | +0.6 | +1.3 | +1.1 |

Table 2: Performance on the temporal action segmentation (TAS) task using EVGAP with AS-Former, C2F-TCN, and LTContext.

| View | Method | verb | object | action | Avg. |
|---|---|---|---|---|---|
| | | **Action Anticipation** | | | |
| Ego | TempAgg | 51.7 | 21.5 | 5.3 | 26.2 |
| | +EVGAP* | 52.5 | 22.1 | 6.0 | 26.9 |
| | +EVGAP | 53.3 | 22.9 | 6.2 | 27.5 |
| | $\Delta$ | +1.6 | +1.4 | +0.9 | +1.3 |
| Exo | TempAgg | 56.8 | 33.2 | 10.2 | 33.4 |
| | +EVGAP* | 57.6 | 33.8 | 10.9 | 34.1 |
| | +EVGAP | 59.2 | 34.9 | 11.5 | 35.2 |
| | $\Delta$ | +2.4 | +1.7 | +1.3 | +1.8 |
| Ego+Exo | TempAgg | 55.1 | 26.9 | 8.9 | 30.3 |
| | +EVGAP* | 56.7 | 27.6 | 9.5 | 31.3 |
| | +EVGAP | 57.4 | 28.1 | 10.3 | 31.9 |
| | $\Delta$ | +2.3 | +1.2 | +1.4 | +1.6 |

Table 3: Performance on the action anticipation task for egocentric-only, exocentric-only, and combined egocentric and exocentric views.

single-view and two-view TAS, respectively, although some metrics decrease. This demonstrates that the visual encoder has a positive impact and can effectively integrate with the downstream model, validating the suitability of the pretrained model. Next, we apply weights from the first step of training using ego-exo video pairs and fine-tune the pretrained model with C2F-TCN, The results are in Table 1 (b), which results in improvements across all performance metrics for both single-view and two-view TAS. This aligns with prior research, which has demonstrated that alignment contributes to more effective feature learning. Furthermore, the performance is also enhanced when only the second pretraining step is applied, shown in Table 1's method (c). We hypothesize that aligning ego and exo videos from multiple viewpoints in the second step not only captures ego-exo relationships but also learns relations in same perspectives (i.e., ego-ego and exo-exo), thereby helping the model learn more comprehensive and dense viewpoint features. When weights from the first step are used to initialize the second step, shown in Table 1's method (d), and the model is pre-trained and subsequently fine-tuned, the performance improves by 0.2% and 0.4%, on average, for single-view and two-view TAS compared to using only the second step. Furthermore, the performance shows an average improvement of 0.9% over the base model for single-view TAS and 1.9% for two-view TAS. These results indicate that the two-step pretraining process effectively enhances the model's ability to capture superior features for both egocentric and exocentric videos. Furthermore, the improvement is greater for two-view TAS compared to single-view TAS, indicating that having a unified feature space for the two-view setting is particularly important for enhancing performance.

## 4.4 STATE-OF-THE-ART PERFORMANCE

To show the effectiveness of the alignment pre-taining, we evaluate with two downstream tasks on Assembly101 to achieve the state-of-the-art performance, i.e., temporal action segmentaion and action anticipation. We apply EVGAP features with fixed or finetuned pre-trained model weight, denoted as '+EVGAP*' and '+EVGAP' in Table 2 and 3.

### 4.4.1 TEMPORAL ACTION SEGMENTATION

For temporal action segmentation (TAS), we choose three models: ASFormer (Yi et al., 2021), C2F-TCN (Singhania et al., 2021), and LTContext (Bahrami et al., 2023). The results demonstrate that using either the fixed weights from EVGAP or finetuned weights for TAS, all metrics are improved, with average increases of 0.8%, and 1.1% for ASFormer, and LTContext, respectively. The fixed EVGAP features directly enhance the performance when training only the downstream model, indicating that our ego-exo video group alignment significantly benefits tasks involving multi-view inputs from both egocentric and exocentric perspectives. Furthermore, fine-tuning the EVGAP features provides greater adaptation to the TAS task, leading to additional performance gains.

| | | Temporal Action Segmentation | | | | | | | | | | | | | |
|---|---|---|---|---|---|---|---|---|---|---|---|---|---|---|---|
| View | Method | F1@{10, 25, 50} | | | Edit | Acc. | Avg. | View | Method | F1@{10, 25, 50} | | | Edit | Acc. | Avg. |
| eg$_1$ | Base | 21.0 | 16.7 | 10.6 | 23.4 | 23.7 | 19.1 | ex$_1$ | Base | 34.7 | 30.1 | 21.8 | 32.3 | 39.4 | 31.6 |
| | +EVGAP | 23.3 | 18.9 | 13.0 | 24.2 | 25.1 | 20.9 | | +EVGAP | 36.9 | 32.8 | 23.3 | 33.4 | 40.6 | 33.4 |
| | Δ | +2.3 | +2.2 | +2.4 | +0.8 | +1.4 | +1.8 | | Δ | +2.2 | +2.7 | +1.5 | +1.1 | +1.2 | +1.8 |
| eg$_2$ | Base | 20.7 | 16.6 | 10.1 | 22.1 | 22.8 | 18.5 | ex$_2$ | Base | 32.8 | 29.0 | 20.5 | 32.2 | 38.0 | 30.5 |
| | +EVGAP | 23.4 | 18.5 | 12.4 | 23.6 | 24.8 | 20.6 | | +EVGAP | 35.7 | 31.4 | 22.7 | 33.7 | 40.3 | 32.8 |
| | Δ | +2.7 | +1.9 | +2.3 | +1.5 | +2.0 | +2.1 | | Δ | +2.9 | +2.4 | +2.2 | +1.5 | +1.7 | +2.3 |

Table 4: Novel view evaluation of the temporal action segmentation task on Assembly101 dataset.

| | | Action Anticipation | | | | | | | | | | |
|---|---|---|---|---|---|---|---|---|---|---|---|---|
| View | Method | verb | object | action | Avg. | View | Method | verb | object | action | Avg. |
| eg$_1$ | Base | 50.9 | 20.3 | 5.1 | 25.4 | ex$_1$ | Base | 57.4 | 32.8 | 11.3 | 33.8 |
| | +EVGAP | 53.8 | 22.8 | 7.1 | 27.9 | | +EVGAP | 60.0 | 35.2 | 13.2 | 36.1 |
| | Δ | +2.9 | +2.6 | +2.0 | +2.5 | | Δ | +2.6 | +2.4 | +1.9 | +2.3 |
| eg$_2$ | Base | 51.2 | 21.8 | 5.9 | 26.3 | ex$_2$ | Base | 56.1 | 32.6 | 11.7 | 33.5 |
| | +EVGAP | 53.5 | 23.6 | 7.5 | 28.2 | | +EVGAP | 58.8 | 34.9 | 13.0 | 35.6 |
| | Δ | +2.3 | +1.8 | +1.6 | +1.9 | | Δ | +2.7 | +2.3 | +1.3 | +2.1 |

Table 5: Novel view evaluation of the action anticipation task on Assembly101 dataset.

### 4.4.2 ACTION ANTICIPATION

For the action anticipation task, we employ TempAgg (Sener et al., 2020) as the downstream model and report the performance on the Ego, Exo, and Ego+Exo splits to evaluate the effect of alignment features across different perspectives. We present the Top-5 recall scores for verb, object, and action anticipation. Additionally, we evaluate using both fixed EVGAP features and fine-tuned EVGAP features, denoted as '+EVGAP*' and '+EVGAP', respectively. The results indicate that the mixed ego-exo data improves by an average of 1.6%, and EVGAP features consistently enhance the performance for both individual ego and exo inputs. This highlights the effectiveness of video group alignment pretraining. Furthermore, the improvement for the exocentric view is 0.5% higher than that for the egocentric view. We hypothesize that it may be due to the imbalance in the pretraining dataset, where the amount of egocentric data is half that of exocentric data, thereby leading the model to preferentially learn more about exocentric videos.

### 4.5 NOVEL VIEW

Following the alignment of the ego-exo video groups, the feature spaces for egocentric and exocentric videos are unified into a common feature space. Consequently, the model is capable of achieving novel view predictions by training on certain viewpoints and testing on others. For temporal action segmentation (TAS) and action anticipation, the results are presented in Tables 4 and 5. The base models for TAS and action anticipation are C2F-TCN and TempAgg, respectively. We select 'eg$_1$', 'eg$_2$', 'ex$_1$', and 'ex$_2$' (1 and 2 mean the first two cameras.) as the novel views in each training setting. Specifically, during model training, one of the views is excluded, and performance is subsequently evaluated specifically on the view. The figures indicate that all metrics improve with the proposed alignment. This demonstrates that mapping both egocentric and exocentric videos to a common feature space shows potential for learning comprehensive view-invariant representation.

## 5 CONCLUSION

In this paper, we propose a new approach for view-invariant representation via video group alignment pre-training. The group video alignment conducts dense contrastive losses over each visual encoder layer. To accommodate more general multi-view data, we perform sparse contrastive learning via egocentric and exocentric video pairs. The two-step pre-training pipeline enables us to realize better performance on various tasks such as temporal action segmentation and action anticipation. In addition, we investigate the zero-shot capacity of the view-invariant model for novel views, where the promising results indicate the potential of learning comprehensive view-invariant representation.

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
