# OpenReview forum: "EVGAP: Egocentric-Exocentric Video Groups Alignment Pre-training"
_ICLR.cc/2025/Conference — ICLR 2025 Conference Withdrawn Submission_

### Official Review · Reviewer_f9HT · 2024-10-22

**Soundness:** 2
**Presentation:** 2
**Contribution:** 2
**Rating:** 3
**Confidence:** 4

**Summary:**

This paper proposes a novel two-step pre-training process for ego-exo video alignment, aimed at improving multi-view video understanding. The approach starts with traditional ego-exo video pair pre-training, followed by an extension to grouped ego-exo videos alignment, allowing the model to capture denser cross-view relationships. The model employs contrastive losses at each layer as auxiliary supervision to enhance training efficiency. The method is evaluated on the Assembly101 dataset, demonstrating its effectiveness in improving downstream tasks like temporal action segmentation and action anticipation.

**Strengths:**

Originality: The paper presents a novel extension of ego-exo video alignment by moving from individual pair alignments to group-based alignments. This shift is supposed to capture richer inter-view relationships, which is a creative and meaningful advance over existing approaches.


Quality: The experimental results are supported by ablation studies, showing improvements on certain evaluation metrics over the chosen base model, but not the state-of-art for the same task, such as ASQuery.


Clarity: The method is clearly described, with well-organized sections explaining the motivation, approach, and results. Figures, such as the visualizations of video alignment processes, effectively illustrate key concepts.


Significance: The contributions have the potential to improve video understanding, particularly for tasks involving multi-view data.

**Weaknesses:**

Data Dependency: The reliance on synchronized ego-exo video pairs imposes a strong constraint on the data, limiting its applicability to datasets where such synchronization is not available.


Mixed Results in Multi-View: The performance of the two-view setting ((d) Base + Step1 + Step2) does not consistently outperform the single-view setting in key metrics (e.g., F1 and Edit scores in Table 1). This contradicts the expectation of multi-view alignment improving overall video understanding. Additionally, it is reasonable to expect that extending the training for more epochs could improve performance in some cases. To ensure a fair comparison, (b) and (c) should be trained for the same total number of epochs or training time as (d). This would demonstrate whether the improvement observed in (d) is due to the proposed method, rather than simply the result of extended training in earlier steps.


Baseline Comparison: The paper compares the proposed method against C2F-TCN, a model not specifically designed for view-invariant features learning. To fully assess the effectiveness of the approach, comparisons with state-of-the-art models in the same domain would provide a stronger baseline, such as the methods listed in section "1. INTRODUCTION"  or a method named ASQuery, which reports better F1@10 performance on Assembly101.


Limited Dataset Coverage: The evaluation is restricted to the limited validation split of Assembly101 dataset, given the model itself trained with the Assembly101 dataset. Including results on additional public datasets like CMU-MMAC, H2O, or Ego-Exo4D would help generalize the findings and demonstrate the robustness of the method across diverse settings.

**Questions:**

In Figure 1 (a), the top labels all describe "the ego video in scene" for four items. Should two of these items be labeled as "exo" videos instead of "ego"?


Section 3.4 discusses the auxiliary loss but does not explain how the values of α and β are selected for each layer during the first and second steps of training. Can you provide more detail on how the weight 0.2, noted in Section 4.2, was chosen?


In formula (1), what is the meaning of the two logarithmic terms? Specifically, what distinguishes the first log from the second in terms of its contribution to the loss?

---

### Official Review · Reviewer_JGNn · 2024-11-03

**Soundness:** 2
**Presentation:** 2
**Contribution:** 2
**Rating:** 5
**Confidence:** 5

**Summary:**

This paper proposes egocentric-exocentric video groups alignment pretraining. The motivation is to align groups of ego and exo videos, in contrast to aligning pairs of ego and exo videos. The authors propose a two-step pretraining strategy and demonstrate improvement on two downstream tasks.

**Strengths:**

+ The motivation is valid. It makes sense to me to explore the idea on utilizing the multi-view videos to form ego and exo video groups (compared with just enforcing ego-exo pair alignment with multi-view videos).

+ The experiments show performance gain of EVGAP. The novel view inference setting (section 4.5) is interesting.

**Weaknesses:**

+ The paper is not very well-written, a few clarifications are needed:
1. L158: How are the scenes defined and identified? Also, is the scene definition incorporated in stage-I training?
2. L161: The objective of EVGAP is to "align and pair" ego-exo video groups. However, this description is overly simplistic. Please expand on what "align and pair" specifically entails in this context. Provide more formal definitions or explanations to clarify the intended meaning.


+ Experiments: evaluation setting is weak. The authors only conduct experiments on two datasets (Charades-Ego and Assembly101) and downstream tasks are only on Assembly101. Moreover, while the authors review a few ego-exo view-invariant works (L37-38), none of them is implemented as a baseline for comparison with EVGAP. The experiments are only about evaluating whether EVGAP gives additional performance gain on top of the task-specific approaches. The gain is expected since EVGAP benefits from more data in the pretraining. However, the real comparison should be about how different ego-exo feature learning approaches perform on these downstream tasks. I believe adding those baselines is essential.


+ Method novelty is limited. I feel the major claim of the paper is to better utilize the multi-view ego-exo videos, extend a regular contrastive loss to account for groups of ego and exo videos. The contribution is incremental from my understanding. Moreover, the claim of using groups and objective 2 is better than using pairs is not thoroughly evaluated. For example, in Table 4 and 5, the authors should report results of aligning all possible views as pairs, to demonstrate the superiority of the proposed objective. Otherwise, the improvement could be understood from introducing multi-view data in pretraining.

Overall, my concern with the paper is its limited novelty and weak evaluation. Specifically, I feel that the biggest claim made in the paper is not thoroughly evaluated, and there is a noticeable absence of baseline comparisons.

**Questions:**

See weaknesses

---

### Official Review · Reviewer_Fg36 · 2024-11-06

**Soundness:** 2
**Presentation:** 3
**Contribution:** 2
**Rating:** 3
**Confidence:** 4

**Summary:**

This manuscript studies leveraging synchronized first-person-view (egocentric) videos and third-person-view (exocentric) videos for self-supervised learning. Specifically, the manuscript proposes to align between a group of ego-videos and a group of exo-video in contrastive learning framework, in contrast to the instance-level contrastive learning.The method is termed EVGAP. EVGAP is developed in two stages, in the first-stage, the standard contrastive loss is applied between paired ego-/exo-videos, in the second stage, the model is then fine-tuned on the group alignment contrastive loss. The pre-training is conducted on Assembly101 and Charades-Ego datasets. Two downstream tasks, temporal action segmentation and temporal action anticipation, are evaluated to show the effectiveness of the proposed methods and ablate the design choices.

**Strengths:**

The paper is structured well and the experiments are executed completely. I found it easy to follow the proposed idea.

**Weaknesses:**

* The proposed method is conceptually simple. The method depends a lot on paired dense egocentric/exocentric videos for the pre-training, which in my opionion is one major drawback of the proposed method. As it is expensive to collect the synchronized videos, so the scalability of proposed method is questionable.
* It is not clearly to me what is the intuition behind contrastively aligning two group of synchronized videos from different view points has benefits over aligning paired video instances from two views.
* In general, the performance difference in main results and in ablation studies are marginal - the difference is less than 2 percent. sometimes comparison between two methods show less than 1 percent difference. For instance in Table 1, the difference on the averaged metric in (c) and (d) are pretty marginal, which does not quite indicate the effectiveness of two stage training design in my view. Considering other factors in the experiments, e.g. having extra layers in proposed model, domain gap in the baseline methods, etc.  more significant results are needed to justify the effectiveness of proposed method and designs.
* Some related works [1][2][3][4] that could leverage ego-exo videos for pre-training under the setup of this manuscript, though discussed in the introduction and related work section, are not included in the baseline results. Would those baselines perform better/worse/equally well on the benchmarks this manuscript studied?
* Overall I think this work is not well-motivated, the proposed method has major limitations and the conducted experiments do not show significant improvements on downstream tasks.

[1] Wang, Q., Zhao, L., Yuan, L., Liu, T. and Peng, X., 2023. Learning from semantic alignment between unpaired multiviews for egocentric video recognition. In Proceedings of the IEEE/CVF International Conference on Computer Vision (pp. 3307-3317).

[2] Xue, Z.S. and Grauman, K., 2023. Learning fine-grained view-invariant representations from unpaired ego-exo videos via temporal alignment. Advances in Neural Information Processing Systems, 36, pp.53688-53710.

[3] Sigurdsson, G.A., Gupta, A., Schmid, C., Farhadi, A. and Alahari, K., 2018. Actor and observer: Joint modeling of first and third-person videos. In proceedings of the IEEE conference on computer vision and pattern recognition (pp. 7396-7404).

[4] Ardeshir, S. and Borji, A., 2018. An exocentric look at egocentric actions and vice versa. Computer Vision and Image Understanding, 171, pp.61-68.

**Questions:**

* Does the number of videos (N) in one group matter for the performance of the proposed method? When N = 1, the proposed method degrades to constrastively train with paired ego-/exo-videos. This seems to be one important baseline to compare against and it is missing from the paper.
* Would un-synchronized video groups work in the proposed framework?

---

### Official Review · Reviewer_4Db2 · 2024-11-06

**Soundness:** 3
**Presentation:** 3
**Contribution:** 2
**Rating:** 5
**Confidence:** 4

**Summary:**

The paper proposes a video pretraining method that aligns a group of ego-centric and exocentric videos. Specifically, it extends previous contrastive learning objective of aligning synchronized video clips of ego-centric and exocentric together by adding multi-view of both ego-centric and exocentric clips into the alignment. A two-stage pretraining strategy is proposed and layer-wise aux losses are added. Experiments are conducted on Assembly101, Charades-Ego datasets and show improvement over baselines.

**Strengths:**

•	Aligning multi views of the same clips is straightforward and the proposed loss is easy to understand.
•	Multiple downstream tasks including temporal action segmentation, action anticipation, and novel view evaluation are considered.

**Weaknesses:**

•	Ablation study did not include the effect of the additional aux losses
•	When comparing to SOTA methods in Table 2 and Table 3, the paper directly build upon previous methods, e.g LTContext, where extra layers and pretraining data are added, so it is unclear where the gain is coming from.
•	The proposed method requires datasets of multi-view aligned videos of both ego-centric and exocentric, which is limited and costly to find.

**Questions:**

•	What is the additional gain of adding the aux losses, can authors give some numbers?
•	The LTContext results in Table 2 is lower than the original paper reported. E.g. LTcontext paper reports 33.9 F1 where in table 2 it is 33.6, can authors give more details about this discrepancy?

---

### Note · Authors · 2024-11-19

I have read and agree with the venue's withdrawal policy on behalf of myself and my co-authors.